# The Co-Evolution of Markets and Regulation in the Japanese Functional Food Industry: Balancing Risk and Benefit

**DOI:** 10.3390/foods14091581

**Published:** 2025-04-30

**Authors:** Keigo Sato, Kota Kodama, Shintaro Sengoku

**Affiliations:** 1School of Pharmacy and Pharmaceutical Sciences, Hoshi University, 2-4-41 Ebara, Shinagawa-ku, Tokyo 142-8501, Japan; satoh.keigo@hoshi.ac.jp (K.S.); kodama.kota@hoshi.ac.jp (K.K.); 2Department of Innovation Science, School of Environment and Society, Institute of Science Tokyo, 3-3-6 Shibaura, Minato-ku, Tokyo 108-0023, Japan

**Keywords:** functional foods, regulatory science, innovation management, risk-side regulation, benefit-side regulation, Japan, Foods with Health Claims, Foods with Function Claims, Good Manufacturing Practices, GMP, co-evolution

## Abstract

Amid the global rise in chronic diseases and escalating costs, functional foods have become a preventive solution, offering benefits beyond essential nutrition. However, the regulatory landscape remains complex, requiring a balance between consumer protection and industry innovation. Previous studies seldom examine how markets and regulation evolve, leaving a clear gap that this review addresses. This study aims to propose a framework focusing on the risk side (quality and safety) and benefit side (functionality) to analyze functional food regulations, and to examine the evolution of regulatory systems and their impact on industry development through a historical analysis of Japan’s functional food regulation from the 1960s. The results reveal that regulations have co-evolved with industry growth, dynamically balancing the risk and benefit sides. Milestones, such as the introduction of the Foods with Function Claims (FFC) system and the Beni-koji (red yeast rice) contamination incident in 2024, highlight adjustments in this balance. The findings suggest that the risk-/benefit-side framework provides a valuable lens for understanding the interplay between regulation and innovation in the functional food industry. This study contributes to regulatory science by offering empirical evidence from the sector and has practical implications for policymakers seeking to design frameworks that promote both innovation and consumer welfare.

## 1. Introduction

### 1.1. Innovation and Regulation in Functional Foods

Amidst the rising tide of chronic diseases and soaring medical costs, expectations for innovation are surging within the medical and healthcare sectors [1,2,3,4,5,6]. Chronic diseases, such as heart disease, cancer, diabetes, and respiratory diseases, are the leading cause of mortality worldwide, accounting for 75% of all deaths globally [7]. Simultaneously, global healthcare spending continues to escalate, reaching 9% of the GDP across Organization for Economic Co-operation and Development (OECD) countries in 2019 [8]. Preventive healthcare, particularly approaches focused on dietary interventions, is gaining prominence for its potential to contribute to disease prevention and extend healthy life expectancy [9]. Beyond general dietary interventions, targeted nutritional strategies are crucial for public health. Food fortification, the practice of adding vitamins and minerals to staple foods, is widely recognized as a highly cost-effective strategy for addressing pervasive micronutrient deficiencies globally [10]. This approach offers demonstrable health, economic, and social benefits, particularly in low- and middle-income countries. However, its implementation requires robust regulatory safeguards to manage risks such as potential over-consumption [10,11].

The functional food market is steadily expanding in response to increasing expectations and food innovation [12,13,14,15,16,17,18]. Globally, the functional food market is a substantial and rapidly expanding segment, valued at nearly USD five hundred billion and consistently projected to grow at high double-digit compound annual growth rates (CAGRs) [19]. This growth is fundamentally propelled by a profound shift in consumer attitudes and priorities, including heightened health consciousness, an ageing global population, a stronger focus on preventive health, and growing interest in personalized nutrition [20]. Consumers increasingly seek products offering specific benefits related to immunity, longevity, cognitive function, and digestive health, while also valuing convenience, clean labels, and plant-based options [19].

The functional food industry consists of not only specialized producers, but a variety of companies, including food companies, pharmaceutical companies, retailers, start-ups using new functional ingredients or local specialty foods, new entrants from other industries, and original equipment manufacturers (OEMs) [16,21].

Focusing on orally ingested products, pharmaceuticals occupy the medical domain, while foods reside in the non-medical domain. Functional foods, distinct from ordinary foods, are characterized by their content of physiologically functional components expected to exert beneficial effects on the body. In this paper, we define ‘functional foods’ as foods that offer health benefits beyond their nutritional value [14,15]. Terms like ‘functional foods’ and ‘nutraceuticals’ are widely used in the market [14,15], sometimes referred to as ‘Nutraceuticals and Functional Foods’ [22] or the ‘Pharma–Nutrition Interface’ [23]. It is important to note that while functional foods offer health benefits beyond basic nutrition, they are fundamentally distinct from pharmaceuticals. Functional foods are regulated as foods and are not intended to diagnose, treat, cure, or prevent any disease, which is the purview of pharmaceuticals. This clear legal and functional boundary is a critical aspect of the regulatory landscape discussed in this paper. Functional foods encompass not only general foods and beverages but also dietary supplements, which share a similar form with pharmaceuticals, such as tablets and capsules. Given their pharmaceutical-like characteristics in form and ingredients, dietary supplements represent a notable segment within the functional food market.

In both medical and healthcare domains, robust regulations ensuring product safety, efficacy, and quality are crucial for consumer protection and market stability [1]. However, the nascent healthcare sector, especially concerning innovative products like functional foods, often lacks well-established regulatory frameworks [6,16,17,24,25]. This necessitates careful institutional design that fosters both consumer safety and industry innovation, balancing regulatory oversight with opportunities for companies to compete and thrive [26,27,28,29,30].

Functional foods have become a subject of academic research in industrial theory, particularly in regulatory science and innovation management. Scholars have explored the interplay of technological and regulatory trajectories in shaping the functional food industry [24,31,32,33,34,35]. Regulations in this area are continuing to evolve, leading to ongoing discussions among regulators, producers, consumers, and academia [32,33,36,37].

These global trends and evolving consumer demands highlight a critical need for innovative solutions in healthcare, positioning functional foods at the forefront. However, the effectiveness and potential of functional foods as a preventive healthcare solution are intrinsically linked to the regulatory frameworks governing their safety, their quality, and the communication of their benefits. Understanding how these regulations interact with market dynamics and foster or hinder innovation in response to global health challenges and consumer needs is therefore of paramount importance. This study responds directly to this need by analysing the co-evolution of regulation and the functional food market in Japan, a leading market with a unique regulatory history.

### 1.2. Objective and Research Questions

Regulation is often noted to have two key dimensions: consumer protection and the promotion of market innovation. The role and impact of functional food regulations have been discussed from various perspectives. However, existing studies on the regulation of functional foods lack a systematic analysis of the balance between the risk side (quality and safety) and the benefit side (functionality and information disclosure). To fill this gap, this study proposes a risk-side/benefit-side framework and analyses the historical evolution of functional food regulation in Japan. Japan is a particularly suitable case due to its large market size and the sequential establishment of a complex regulatory system [16,17,18]. In doing so, we contribute to the theoretical development of regulatory science and innovation management and provide practical suggestions for policymakers and industry players.

Specifically, this study addresses the following research questions:How have Japanese functional food regulations evolved from a risk-and-benefit perspective?How have these regulations affected and been affected by the development of the industry?How should the design of functional food regulations take into consideration interactions with regulations in adjacent industries?

To answer these questions, we conduct a literature review to establish the theoretical foundation of the risk-side/benefit-side framework and analyse the historical evolution of Japan’s functional food regulations. This study aims to contribute to regulatory science and advance the theory of industry convergence.

### 1.3. Japan’s Functional Food Regulations

In Japan, the ‘Foods with Health Claims (FHC)’ system has been established as a regulatory system for functional foods. In the market, products that comply with the FHC system and those that do not coexist. The functional food industry straddles the boundaries of the FHC system and is characterized by its larger scale outside the system. Furthermore, the FHC system consists of three segments. Table 1 provides a comparative overview of Japan’s multi-tiered system for functional foods, distinguishing between ‘so-called health foods’, the FHC system, including Foods with Function Claims (FFCs) and Foods for Specified Health Uses (FOSHUs), and pharmaceuticals. It highlights key differences in the administrative process (notification vs. approval), the required basis for efficacy and safety, and the status of GMP implementation across these categories. Notably, Table 1 also illustrates the significant difference in market scale among these segments and shows the rapid expansion of the FFC market from JPY 31 billion in 2015 to JPY 357 billion in 2020, surpassing the FOSHU market size of JPY 321 billion in the same year [38]. Compared with pharmaceutical regulations, in terms of quality, safety, and efficacy (functionality), dietary supplements especially incorporate regulations based on pharmaceutical concepts, such as Good Manufacturing Practice (GMP) and clinical trials. Among the four categories of dietary supplements, the contents and levels of regulations are different from each other. Between dietary supplements and pharmaceuticals, there are some similarities in terms of quality, safety, and functionality (or efficacy), such as GMP and clinical trials. Compliance costs and the potential for product differentiation, which can act as an incentive for regulatory compliance, vary among the four categories of dietary supplements, depending on the level of regulation. The regulatory complexity of dietary supplements offers companies strategic options for responding to regulations.

Any functional food produced overseas can circulate in Japan only after it passes a border inspection conducted by the quarantine offices and is filed through an Import Notification for Foods following the Food Sanitation Act [39]. The filer must document raw-material provenance, manufacturing controls, and evidence of safety and functionality identical to those required for domestic products. If, however, the overseas product (i) contains an ingredient that is classified as an active pharmaceutical substance in Japan or (ii) displays wording that implies a therapeutic claim, the item is legally re-classified as a drug under the Act on Securing Quality, Efficacy and Safety of Pharmaceuticals and Medical Devices. In this case, the importer must obtain a pharmaceutical manufacturing and marketing license and secure product-by-product approval before any commercial distribution, and the importation of the product as a ‘food’ is prohibited. These additional hurdles create a de facto regulatory divide between domestic and non-domestic functional foods, anchored in traceability requirements and the Japanese definition of medicinal efficacy. From an innovation science perspective, this de facto separation, while potentially offering a degree of protection to domestic industries by leveraging national regulatory definitions and traceability, also presents challenges for international regulatory harmonization and market access for foreign companies, influencing the global dynamics of the functional food sector.

### 1.4. The Structure of This Study

The structure of this paper is as follows: Section 2 provides a literature review on regulatory theory and the current state of functional food regulation. Section 3 analyses the historical evolution of functional food regulation in Japan. Section 4 discusses the relationship between regulation and industry development using a risk-side/benefit-side framework and some implications. Finally, Section 5 presents conclusions.

### 1.5. Materials and Methods

This study combines a narrative literature review with a time-series analysis of publicly available regulatory data. First, peer-reviewed sources were retrieved from Google Scholar and complementary databases using search strings that paired functional food with terms such as regulation, policy, innovation, risk management and health claims. Relevant articles were screened in full text, and additional records were gathered through backwards and forward snowball sampling. Second, longitudinal regulatory data were assembled from administrative documents and industry white papers. Official texts—including Cabinet Office ordinances, Consumer Affairs Agency (CAA) guidelines, and Ministry of Health, Labour and Welfare (MHLW) notices—were downloaded from ministerial websites and government repositories. Each document was read in detail to identify discrete regulatory amendments over time. Every amendment was then coded as either a tightening or loosening measure and classified along two analytical axes: the risk side (quality–safety provisions) and the benefit side (functionality–claim provisions). This dual coding enabled a chronological map of how Japanese functional food regulation has shifted its emphasis between consumer protection and innovation incentives.

## 2. Literature Review

This section is structured as follows: First, Section 2.1 discusses the role of regulation in shaping the convergence process, from the viewpoint of the functional food industry being a result of convergence. Next, Section 2.2 presents the findings of the literature review on functional food regulation itself, beginning with a comparative overview of regional regulatory approaches, highlighting the diversity of regulations across different countries (Section 2.2.1), followed by an examination of key regulatory dimensions, including safety, quality, and the regulation of health claims (Section 2.2.2). Section 2.3 synthesises the reviewed literature and identifies the research gap that this study aims to address. Section 2.4 then introduces the proposed risk-side/benefit-side framework, which systematically analyses the dual dimensions of functional food regulation, and Section 2.5 positions this framework within existing regulatory theories.

### 2.1. Regulation in the Convergence Industry

In industry theory, functional foods have been discussed as an industry resulting from the industrial convergence of pharmaceuticals and foods [24,31,32,33,40,41]. Industry convergence is a blurring of industries distinguished by products, actors, knowledge and learning processes, technologies, inputs, demand structures, competition, and processes (standards or regulations). Industry convergence blurs existing industry boundaries and is also an opportunity for innovation because it plays an important role in the formation of new markets and industries. Industry convergence studies have discussed various influences on functional foods from the pharmaceutical and food industries. Specifically, it is suggested that technological competence is introduced from the pharmaceutical industry and marketing competence from the food industry [32,42].

Industry convergence progresses through multiple stages, such as knowledge (or science), technology, application (or market), and industry [34,35,43]. Since all convergence processes blur the boundaries between industries, firms are faced with new technologies, consumers, and needs. Industry convergence can be categorized as supply-side-originating and demand-side-originating [43]. While convergence on the supply side is mainly due to technological innovation originating from technology and knowledge, convergence on the demand side is driven by changing consumer needs and regulatory changes [34]. Convergence by regulation occurs at a faster rate than that caused by changing needs, and its speed is affected by the establishment or relaxation of regulations [44,45]. Deregulation is the driving force of convergence on the demand side and leads to the entry of firms without highly technological development [45,46,47,48,49]. Some cases, such as the environmental industry, have been identified as those where knowledge has been fused to respond to regulations, and technological development has progressed [44].

Figure 1 conceptually illustrates the dynamic landscape of regulatory and industry convergence among non-medical (food), healthcare (functional food), and medical (pharmaceutical) sectors. As depicted, these domains are arranged along a continuum, with the vertical axis indicating the level of regulatory stringency, being highest in the medical domain and lowest in the non-medical domain. The healthcare domain, where functional foods reside, is positioned between these, representing a dynamic space where both industry structures and regulatory frameworks are undergoing convergence and mutual influence. Regulatory convergence involves increasing similarity and the blurring of boundaries between regulations in adjacent domains, driven by evolving science and market needs, while industry convergence entails the fusion of products, technologies, and actors. Effectively regulating this converging healthcare domain presents a critical challenge, as visually represented by the risks of ‘Over-strict: Regulatory Failure’ (leading to innovation barriers) and ‘Over-weak: Market Failure’ (resulting in quality and safety issues). Successfully navigating this complex landscape, finding and maintaining an appropriate regulatory balance that avoids these failures and fosters the dynamic interplay between regulation and industry, is crucial for fostering both consumer welfare and innovation within the functional food sector.

### 2.2. Findings from the Literature Review on Functional Food Regulation

From the viewpoint of preventing market failure, regulations in the medical and healthcare sectors, including functional foods, are essential. Regulations in this sector address various aspects arising from the characteristics of the products and market [50].

#### 2.2.1. Comparative Overview of Regional Regulatory Approaches

Regulations regarding functional foods are diverse and vary across different countries. In the European Union (EU), the European Commission serves as the main regulatory authority, while other regions have agencies like the Food and Drug Administration (FDA) in the United States, the Ministry of Food and Drug Safety (MFDS) in Korea, the Taiwan Food and Drug Administration (TFDA), the Chinese Food and Drug Administration, the Health Sciences Authority (HAS) in Singapore, and the Consumer Affairs Agency (CAA) in Japan. Each country establishes its own regulations and guidelines for functional foods [25,50,51,52].

Functional foods are generally treated as a distinct category from pharmaceuticals, with specific laws and regulations. Labelling requirements differ internationally, with some countries mandating the specification of product forms, such as tablets or capsules, on packaging. Table 2 shows the administrative process, whether or not a GMP system for dietary supplements is required, and whether or not it can be used as evidence in clinical trials. It highlights Japan’s uniquely multi-track system (FOSHUs, FNFCs, FFCs, plus unregulated ‘so-called’ health foods). No other country combines an approval track, a notification track, and an entirely voluntary track within one market. In the U.S., which is the largest market for supplements, the Dietary Supplement Health and Education Act (DSHEA) allowed dietary supplements to be labelled with functional claims as of 1994. Since then, the dietary supplement market in the U.S. has expanded significantly. Labels related to functionality increase consumer willingness to purchase and stimulate the market. However, the DSHEA, applicable only to dietary supplements, does not require clinical trials of individual products. In addition to the U.S., according to Singapore’s regulations, which require notification, clinical trials for individual products are not required; however, regulations in some countries, such as South Korea and Taiwan, require approval for individual products [25]. Functional foods are typically marketed to enhance health as supplements or nutrient sources, rather than as treatments for diseases, with marketing them as medical treatments generally prohibited. To ensure safety and quality, manufacturers often implement GMP systems, which also vary by country, with dedicated GMP systems for dietary supplements in the USA, Korea, Taiwan, and China, while Japan and Singapore do not mandate specific GMPs. Regulatory bodies often maintain ‘positive lists’ of permissible ingredients to assure safety and may also utilize ‘negative lists’ of prohibited ingredients. Intellectual property for functional foods is also addressed in some regions, such as Japan, where patent standards have been revised to protect novel attributes of known foods.

Although the fortification of staple foods and condiments is promoted worldwide as a cost-effective public health strategy, numerous operational, regulatory, and consumer-acceptance challenges remain before large-scale programs can achieve their intended impact [53]. In Japan, any food that is enriched with vitamins or minerals is regulated as a Food with Nutrient Function Claims (FNFC) provided that the added nutrient falls within the compositional ranges listed in the CAA Standard Specification Table and that the product otherwise complies with the Food Sanitation Act. Through this self-certification pathway, manufacturers may display standardized wording, thereby lowering entry barriers while still anchoring the claim in a science-based specification.

Overall, regulations governing functional foods encompass aspects such as product categorization, shape description, product purpose, ingredient lists, GMP systems, and intellectual property protection. These regulations, similar to those for pharmaceuticals, aim to ensure quality, safety, functionality (efficacy), information disclosure, and labelling [13,14,54,55,56,57,58,59]. These regulatory goals are established within the context of information asymmetry [60] between consumers and producers, where consumers typically have less information regarding the quality, safety, and functionality of functional foods. Fundamentally, functional food regulations are designed to protect consumers and foster fair market competition [12,21,61,62,63].

#### 2.2.2. Key Regulatory Dimensions: Safety, Quality, and Health Claims

Drawing from the literature review, several key regulatory dimensions emerge in the global functional food landscape. These dimensions primarily address information asymmetry regarding product attributes and aim to protect consumers while influencing market dynamics.

Consumer Protection from Quality Issues

One primary area of concern in functional food regulation is consumer protection from quality issues. Information asymmetry regarding quality presents a safety risk, particularly with dietary supplements, as consumers cannot easily assess product quality by appearance. Factors such as ingredient quality, ingredient interactions, and impurities significantly affect product integrity. Allergen disclosure (e.g., soy, egg) is mandated under the Food Labelling Act in Japan and forms part of risk-side regulation. Given the concentrated nature of dietary supplements, quality defects can pose substantial safety risks. Historical issues in the functional food market, such as poor manufacturing process control and insufficient ingredient levels, underscore this concern [64,65,66]. Beyond these, instances of outright fraud, including exaggerated claims lacking scientific backing, inaccurate ingredient labelling, and contamination have periodically surfaced, further highlighting the critical need for robust risk-side regulation to safeguard public health and consumer trust [67,68,69].

Quality regulations and quality certifications are, therefore, crucial for assuring safe and high-quality products and building consumer trust through adherence to manufacturing standards like GMP [16,56]. While compliance with quality regulations involves costs for companies [70,71], it demonstrably improves product quality [59], reduces complaints, enhances trust [56], and can yield internal benefits like increased productivity [56] and external benefits such as improved customer relations [72,73,74]. Review findings indicate that the stringency and mandatory nature of these quality control measures, including GMP for dietary supplements, vary significantly across regions, reflecting the diverse regulatory approaches discussed in Section 2.2.1. Ensuring consumer protection from quality issues thus remains a central objective of functional food regulation globally.

Information about Health Benefits

Another significant area is ensuring consumer access to reliable information about both safety and health benefits. The subtle effects of functional ingredients can make it difficult for consumers to accurately evaluate potential health benefits. Despite strong consumer interest in benefit information [75,76,77], information asymmetry hinders informed decision-making. In today’s media-saturated marketplace, dominated by social networks, e-commerce platforms, ‘fake-news’ outlets, and poorly vetted popular science products ranging from specialized clothing and footwear to fad diets, dietary supplements and training programs are routinely marketed with unsubstantiated promises of instant cures or dramatic health improvements [78]. Unreliable or misleading information can impede consumer choice and erode market trust. Consequently, information on health benefits and product functionality is regulated through health claims and labelling requirements. These regulations aim to standardize the communication of benefits and ensure that claims are substantiated and not misleading. The literature reveals that regulators globally employ diverse systems for managing health claims.

Obtaining health claim approvals involves costs for companies, including R&D to validate claims and administrative processes [79]. However, approved health claims enhance product value and can incentivize innovation by providing a framework for companies to differentiate their products based on substantiated benefits and reduce unfair competition from unsubstantiated claims [17,18,63,75,76,77]. Conversely, overly strict regulations on health claims can also increase product development costs, potentially inhibiting innovation and market competition, especially for smaller firms with limited R&D resources [79].

Administrative processing costs are affected by the kind of process used to obtain health claim approval. Such processes fall into several categories, including notification, approval, etc. [17,25,63,79]. Studies have discussed the gains and losses of the approval system and generic labelling (by notification and other methods). In the method that allows unique labelling under the approval system, a single company can produce multiple products with unique labelling, thereby increasing consumer awareness. The downside is higher out-of-pocket expenses for both the firm (clinical trials, dossiers) and the regulator (expert review), as well as a slower time to market, especially problematic for claims that consumers value less highly. Meanwhile, generic labelling could allow multiple companies to launch similar products, so that consumers would have more opportunities to be exposed to the labelling of functionality [63,79].

Review findings demonstrate significant divergence in how regions manage health claims (as detailed in Section 2.2.1), with varying requirements for scientific evidence and administrative processes.

Relatively weak regulations requiring a low level of evidence would make it easy for companies to enter the market [80]. However, in an inadequate regulatory environment, low-quality products produced at low cost may be introduced into the market, and adverse selection may result in these products occupying the market. It is noted that healthy market competition requires appropriate regulations for product health claims [61]. The appropriate regulation of functional foods is needed from the perspectives of consumer protection and market competition [62,63,81].

The literature suggests that balancing consumer protection from misleading information with fostering innovation through clear claim pathways is a persistent challenge addressed differently across regional regulatory frameworks.

### 2.3. Synthesis of the Literature and Research Gap

To summarize the above, the aspects of functional food regulation are organized in Table 3. Existing studies lack a systematic analysis of the balance between the risk and benefit sides. In particular, studies examining the impact of regulation on industrial development from both sides are limited. The objective of this study is to fill this gap.

### 2.4. Risk-Side/Benefit-Side Framework to Analyse Functional Food Regulation

Based on the above review of regulatory aims and practices in the functional food sector, it becomes evident that functional food regulations inherently address two fundamental, intertwined dimensions: risk-side concerns related to quality and safety, and benefit-side concerns related to functionality and consumer information. Building upon the regulatory challenges and aims identified in the healthcare and functional food sectors, this study proposes the risk-side/benefit-side framework to systematically analyse these dual dimensions of functional food regulation.

Risk-side regulation: This dimension centres on consumer protection by safeguarding the quality and safety of functional foods. It aims to minimize market failures stemming from externalities, ethical considerations, and information asymmetry related to quality and safety. Core regulatory tools within risk-side regulation include GMP, quality certification schemes, safety standards, and pre-market safety assessments. Addressing risk-side concerns is a fundamental goal of functional food regulation, particularly in sectors impacting public health [82,83,84].Benefit-side regulation: This dimension focuses on facilitating informed consumer choice by ensuring the reliable communication of product benefits, primarily functionality. It aims to address information asymmetry concerning product benefits. Key regulatory instruments within benefit-side regulation include health claim systems, labelling regulations, and requirements for the scientific substantiation of functionality claims. Benefit-side regulation plays a crucial role in promoting market efficiency and empowering consumer choice by providing credible information on product benefits [63,77].

The risk-side/benefit-side framework emphasizes that effective functional food regulation necessitates a balanced approach that adequately addresses both dimensions. An imbalance, with excessive focus on either risk-side or benefit-side regulation at the expense of the other, may lead to suboptimal outcomes. Neglecting risk-side regulation can result in market failures, erode consumer trust, and potentially cause harm [63]. Conversely, neglecting benefit-side regulation can stifle innovation and limit consumer access to potentially beneficial products, potentially hindering market development and innovation [70,71]. The optimal regulatory strategy seeks to harmonize risk-side and benefit-side considerations, fostering an environment that simultaneously assures robust consumer protection regarding quality and safety (risk-side) and encourages innovation in product functionality through well-designed benefit-side regulation, recognizing that both dimensions are crucial for a well-functioning and beneficial functional food market.

Beyond a static balance, the framework highlights the potential for a dynamic interaction where actions within one dimension positively reinforce the other, creating a virtuous cycle. Figure 2 conceptually illustrates a key mechanism of this dynamic interplay within the functional food sector, involving both regulatory influences and market dynamics across company, product, and consumer levels.

As shown in Figure 2, the process is sometimes initiated by tightening risk-side regulation, which directly leads to firm costs and generates a compliance incentive. This incentive, coupled with investments (reflected in firm costs) in robust quality control and safety measures, such as adhering to GMP, enhances product credibility [16]. On the benefit side, the use of health claims is a primary driver of consumer demand. The desire to effectively leverage health claims in the market also provides a strong compliance incentive for firms to adhere to underlying regulations [18]. Consumer demand, in turn, serves as a critical signal back to firms, stimulating Research and Development (R&D) to develop new products and functional ingredients. This R&D then fuels the use of health claims by providing new, scientifically substantiated benefits to the market [17]. Ultimately, both enhanced product credibility (driven by risk management and compliance) and increased consumer demand (driven by effective claims and product appeal) contribute to market growth. This dynamic interaction illustrates how strategically linking risk-side requirements with benefit-side opportunities can create a self-reinforcing cycle, driving both improved safety/quality and continued innovation in the functional food market, reflecting the co-evolution of regulation and industry.

### 2.5. Theoretical Positioning of the Framework Compared with Existing Frameworks

The risk-side dimension could be linked to the concept of social regulation, aiming to safeguard public health and consumer safety, and the benefit-side dimension could be linked to economic regulation focused on addressing information asymmetry and stimulating market efficiency. While existing regulatory theories, as outlined below, including those addressing social and economic regulations and innovation-focused perspectives, do not explicitly employ this dichotomy, our framework reveals their implicit relevance to both benefit-side and risk-side objectives. Positioning this dual-focus regulatory framework within the broader landscape of social regulation and economic regulation clarifies why balancing protection from market failure and innovation incentives is especially critical in healthcare convergence industries like functional foods.

#### 2.5.1. The Porter Hypothesis

The Porter Hypothesis proposes that well-designed environmental regulations can stimulate innovation and enhance competitiveness by encouraging firms to develop new technologies and processes [27]. This hypothesis emphasizes that regulations, while potentially increasing costs, also provide incentives for firms to innovate. Similarly, the risk-side/benefit-side framework focuses on how regulations can incentivize firms to develop new technologies and processes, particularly in the healthcare sector. However, the risk-side/benefit-side framework extends beyond the Porter Hypothesis by explicitly considering both the risk-side and benefit-side dimensions of regulation. This dual focus allows for a comprehensive analysis of how regulations impact both consumer protection and market innovation, offering unique value in the context of healthcare convergence industries like functional foods.

#### 2.5.2. Information Asymmetry Theory

Information asymmetry theory emphasizes the challenges posed by unequal information between consumers and producers, which can lead to market failures such as adverse selection and moral hazards [60]. The risk-side/benefit-side framework directly addresses these challenges through benefit-side regulations, such as health claim systems and labelling requirements, which aim to reduce information asymmetry and empower consumers to make informed choices. By integrating risk-side and benefit-side considerations, the framework provides a more holistic approach to mitigating information asymmetry in the functional food market.

#### 2.5.3. Regulatory Capture Theory

Regulatory capture theory highlights the risk that regulatory agencies may prioritize industry interests over consumer protection, leading to suboptimal outcomes [85]. While this theory underscores the potential for regulatory failure, the risk-side/benefit-side framework provides a balanced approach to mitigate such risks. By explicitly addressing both risk-side and benefit-side dimensions, the framework ensures that regulations are designed to protect consumers while fostering industry innovation. This dual focus reduces the likelihood of regulatory capture by maintaining a balance between consumer protection and industry incentives.

## 3. History of and Changes in the Functional Food Regulatory System

This section presents a detailed history of Japan’s regulatory system for functional foods. The system has undergone a historical transition.

Regulatory approaches to risk management in areas like food and health often differ significantly across jurisdictions, particularly concerning the application of the precautionary principle. The precautionary principle dictates taking preventive action when there is scientific uncertainty about potential risks, opting for a safety-oriented approach. The European Union explicitly enshrines this principle in its fundamental food law, the General Food Law, and actively applies it in areas such as genetically modified organisms (GMOs) and food additives [52]. In contrast, the U.S. (Food and Drug Administration: FDA) generally adopts a risk assessment-centred approach, prioritizing the demonstration of harm based on available scientific evidence [51]. Japan does not explicitly adopt the precautionary principle; instead, risk assessment is the responsibility of the Food Safety Commission, while risk management is shared among various ministries, including the Ministry of Health, Labour and Welfare, the Ministry of Agriculture, Forestry and Fisheries, and the Consumer Affairs Agency. The later-discussed Beni-koji incident highlights challenges in food safety and has prompted discussions regarding preventative mechanisms, reflecting the ongoing relevance of this debate.

This difference in approach often manifests in situations where scientific data on potential long-term health risks is limited or inconclusive. While a risk assessment-centric approach prioritizes evidence of harm, the precautionary principle emphasizes taking preventive measures even in the absence of full scientific certainty, reflecting different societal tolerances for risk and uncertainty in the context of public health. The challenge lies in balancing the need for innovation with the imperative to protect consumers from potentially unknown or long-term hazards.

To analyse the transition in Japan’s functional food regulations, we conducted a historical analysis spanning from around the 1960s to 2025. The year 1971 is likely the starting point of modern functional food regulation discussions in Japan, because in this year (Showa 46 in the Japanese calendar), the ‘46 Notice’ was published. Our analysis focused primarily on the FHC system as a representative example of benefit-side regulation and GMP guidelines as a key example of risk-side regulation.

### 3.1. Methodology

Data collection relied on publicly available information, encompassing administrative documents and industry association materials. Administrative documents were sourced from the official websites and reports of governmental bodies, notably the CAA and MHLW.

The collected administrative documents were subjected to a content analysis. This analysis involved reading the documents to identify specific changes in the regulations over time. Furthermore, each identified regulatory change was categorized based on whether it represented a strengthening or weakening of regulatory control. This categorization was conducted through the lens of risk-side and benefit-side perspectives, with GMP guideline changes primarily assessed for their impact on risk-side aspects (e.g., consumer safety, quality assurance) and FHC system changes primarily assessed for their impact on benefit-side aspects (e.g., market access, industry promotion).

By tracing these regulatory shifts and categorizing them within the risk-side/benefit-side framework, we aim to understand how the balance between these two aspects has evolved in Japan’s functional food regulation and its relationship with industry development.

This study uses only publicly available information and does not include any information that constitutes personal information or trade secrets.

### 3.2. Description of the Regulatory Change in Functional Foods

#### 3.2.1. Before 1960: No Concept of Functional Food

Legally, food has no function in preventing or treating disease, and this is the clear boundary between food and pharmaceuticals. Before the advent of dietary supplements, food in the form of tablets or capsules did not exist in the first place. In the 1960s, products in the form of tablets or capsules were only pharmaceuticals.

#### 3.2.2. 1960s: Rise of Functional Food Market

In the 1960s, the health boom arrived in Japan, and the concept of ‘health food’ emerged. But the lack of a rule for healthy food brought a state of anarchy and health hazards [86].

#### 3.2.3. 1970s: Tightening of Regulations on Form

In 1971, the Pharmaceutical Affairs Bureau of the Ministry of Health and Welfare issued the ‘Standards on the Scope of Drugs’ (so-called ‘46 Notice’, named after the 46th year of Showa), which notified prefectural governors to strengthen the crackdown on food products in pharmaceutical-like forms, such as tablets or capsules, as these fell under the category of unapproved unlicensed drugs [87]. In other words, ‘dietary supplements’ were treated as a violation of the Pharmaceutical Affairs Law. In reality, however, ‘dietary supplements’ existed in a grey area. For example, supplements were sold in the form of triangular or other special-shaped tablets instead of round tablets, which were considered to be pharmaceutical shapes.

#### 3.2.4. 1980s–Early 1990s

This period saw the development of research on food functionality and the establishment of the Food with Health Claims system (Foods for Special Health Uses: FOSHUs).

In the 1980s, research on the functional properties of foods began to progress. In Japan, the ‘Systematic Analysis and Development of Functional Foods’ (1984) by the Ministry of Education, Culture, Sports, Science and Technology, and the ‘Functional Food Study Group’ (1988–1990) by the Consumer Health Bureau of the Ministry of Health and Welfare were conducted. Based on the results of food functionality research, the concept of food functionality was established as a legal system in 1991 with the establishment of the FOSHU system. However, only obvious food forms were allowed for FOSHU. The FOSHU system was recognized as a pioneering system and was featured in *Nature*, leading to the creation of a JPY-600-billion market by 2005 [88].

In Japan, functional foods were institutionalized by the FOSHU system in 1991. The markets for functional foods, including FOSHUs and Foods with Nutrient Function Claims (FNFCs) were small compared to those outside the regulation, which were larger.

#### 3.2.5. Late 1990s: Relaxation of Regulations on Form

In the late 1990s, there was a request for deregulation from the U.S. in response to the so-called ‘46 Notice’ regulation on dietary supplements [89]. The U.S. government and the American Chamber of Commerce in Japan raised the issue and brought it to the Cabinet Office’s ‘Market Opening Issues Complaint Handling System’ in 1996. The ban on vitamins, herbs, and mineral supplements was lifted in 1997–1999.

In the Report of the Office of Trade and Investment Ombudsman (OTO), it was decided to review the scope of pharmaceutical products concerning vitamins and other substances. Subsequently, the ‘46 Notice’ was amended, and the restrictions on the shape of tablets or capsules were removed in 2000 [90].

#### 3.2.6. 2000s: Revision of Food with Health Claims System

In 2001, the Foods with Health Claims system was revised, allowing the inclusion of tablets and capsules in FOSHU and creating a new category of Foods with Nutrient Function Claims. Under this self-certification system, foods or tablets and capsules containing certain vitamins, minerals, and other nutrients at certain content levels can be labelled as Foods with Nutrient Function Claims [90].

GMP has been discussed since its introduction in Japan in the 2000s, after it was introduced in the U.S. by the DSHEA in 1994.

In 2005, the MHLW of Japan issued ‘GMP Guidelines’ requiring the industry to work to ensure the safety of dietary supplements.

This provided GMP management guidelines from the receipt of raw materials to the packaging and shipment of the final product. In response, two organizations in the industry began voluntary GMP certification. In 2014, the Health Food Certification Council designated two organizations working on certification since 2005 as certification bodies [16].

#### 3.2.7. After 2015: Establishment of Foods with Function Claims (FFC) System

In 2013, the Council for Regulatory Reform and the Japan Revitalization Strategy, which aim to contribute to national growth and development, the stabilization of and improvement in people’s lives, and the revitalization of economic activities, identified the development of a system to enable the functional labeling of general health foods as a relevant issue and started discussions on regulatory reform in the functional food field [91,92,93]. As an issue in the existing regulation, the FOSHU system, it was pointed out that the high cost of regulatory compliance was a barrier to entry for small- and medium-sized enterprises. The high cost was due to clinical trials of safety and efficacy, which were required for each food product, and the approval process was time-consuming and costly. As part of regulatory reform, it was decided that the labelling system for dietary supplements in the U.S. would be used as a reference, to utilize the know-how of the private sector by having companies evaluate the scientific basis for labelling and then label the function of the product themselves [91].

During 2013–2014, the ‘Study Group on a New Functional Labeling System for Foods’ met eight times to discuss the direction of the new system in terms of ensuring safety, scientific grounds, labelling, and the government’s involvement [93]. The study group consisted of academics in the fields of food science and pharmacology, physicians, consumer groups, media, and industry associations related to healthy foods. Through these discussions, the new system also reflected the opinions of the industry [92]. The Food Labeling Law was subsequently revised, and the FFC system was established in 2015 [55,56]. The system design for functional claims was based on the DSHEA in the U.S., but did not include the concept of form regulation. The system spans both obvious food forms and dietary supplement forms, such as tablets and capsules. In the case of the DSHEA, after its 1994 passage, strong congressional and industry pressure pushed the FDA into a reactive stance, intervening only when clear evidence of harm emerged [94]. This approach contrasts sharply with Japan’s more precautionary regulatory philosophy.

Under the FFC system, the functionality of a product can be included in a label by submitting a notification, which is a simpler administrative procedure. In addition to ‘clinical trials on the product’, a systematic review (SR) of the publicly known literature on the ingredient became acceptable as the basis for the functional claim. The FFC system is a deregulation against the FOSHU system, as it allows for the launch of products that conform to the FHC system at a lower cost than the FOSHU system [17]. On the other hand, FFC also includes the aspect of tightening regulations on quality control [16]. The notification guidelines for FFCs [55] published by the CAA stated that ‘for processed foods in the form of dietary supplements, manufacturing process control based on GMP is highly desirable’. This means that the CAA effectively made GMP mandatory for dietary supplements under the FFC system. Since the establishment of the FFC system in 2015, the market for functional foods under FFC regulation has expanded rapidly [17,18]. Specifically, as shown in Table 1, the FFC market expanded rapidly from JPY 31 billion in 2015 to JPY 357 billion in 2020, surpassing the FOSHU market size of JPY 321 billion in the same year [38]. Regulatory changes in Japan have significantly impacted functional foods.

#### 3.2.8. 2024: Recent Health Incident of Dietary Supplements and Regulatory Response

From late 2023 to early 2024, a significant health concern emerged in Japan, specifically related to FFC dietary supplements containing Beni-koji (red yeast rice) [95,96,97,98,99]. Reports of kidney dysfunction and fatalities prompted the manufacturer to announce a voluntary recall on 22 March 2024. Soon after, the MHLW classified the implicated products as violations of the Food Sanitation Act and ordered their disposal. As a preventive measure, authorities directed 225 companies associated with the raw material to conduct self-inspections, though none reported further contaminated products. In parallel, the CAA promptly investigated all of the approximately 7000 FFC products containing ingredients other than red yeast rice and consolidated reports of health concerns [100].

According to the government’s findings, the presumed cause of this incident was the contamination of red yeast rice by a blue mould (*Penicillium adametzioides*) during the cultivation stage, producing toxic puberulic acid that led to kidney disorders [101,102,103]. It was also noted that the company waited nearly two months after becoming aware of the harm before notifying administrative authorities and that delays in consumer warnings and product recalls contributed to further damage [98,104].

This contamination incident also impacted FFC regulations. In April 2024, soon after the recall announcement, the government established a ‘Study Group on Foods with Function Claims’, which held six meetings over two months and, at an exceptionally rapid pace, issued a report by May 2024 [105,106]. The group focused on safety measures, including mandatory notification to administrative agencies of any health incidents, stricter manufacturing and quality control for supplement-type foods, and clearer consumer information regarding the proper use of FFCs.

In response, on 1 September 2024, new regulations concerning the notification of FFCs were enacted into law, introducing the following strengthened measures: the implementation of a health damage reporting system, provision of health damage information to health centers based on medical diagnoses, mandatory GMP for the manufacturing and processing of certain dietary supplement-type foods using natural extracts, and revised notification labeling methods on product packages [100,107]. While the reporting system took effect immediately, other measures will be phased in with a grace period. These regulations are legally binding, placing an ‘obligation’ on notifiers to comply. Moreover, the former ‘guidelines’ for FFC notification have been rebranded as a ‘manual’.

## 4. Functional Food Regulatory Schemes for Industrial Systems Formation

This section is structured as follows: First, Section 4.1 examines the complexity of Japan’s regulatory system’s historical evolution and highlights the dynamic balance between risk-side and benefit-side regulations. Section 4.2 explores how the FFC system has driven industry development by balancing these two dimensions. Section 4.3 reflects on the contamination incident as a critical lens for understanding the interplay between regulatory frameworks, industry practices, and consumer safety. Section 4.4 analyses the interactions between functional food regulations and related industries, emphasizing the role of regulatory convergence in shaping the broader industrial ecosystem. Finally, Section 4.5 discusses the prospects for global markets and regulations. The section concludes with implications and future research in Section 4.6 and Section 4.7.

### 4.1. Complexity of Regulatory Systems in Japan and Development of Industry

Table 4 illustrates the regulatory changes on the risk side (safety or quality control) and the benefit side (health claims) in functional foods. Regulation has emerged alongside industrial development, progressing through phases of relaxation and strengthening. Japan’s historical trajectory demonstrates the complexity of navigating the healthcare convergence landscape (as conceptually depicted in Figure 1), involving a dynamic balancing between risk-side and benefit-side elements. Notably, a balance between risk-side and benefit-side elements is discernible in these regulatory adjustments. Figure 2 illustrates a key mechanism of this dynamic interplay, showing how these dimensions can interact to drive market and compliance outcomes over time. The rest of this section describes the historical transition in functional food regulation in Japan from the perspective of the risk side and the benefit side of regulation.

Historically, Japan has had a cultural interest in the health impacts of food. When dietary supplements initially entered the market, they operated in an environment devoid of specific regulations. This nascent stage of the functional food industry was characterized by a lack of established regulatory frameworks. Subsequently, as the dietary supplement market grew, dietary supplements in pharmaceutical forms were banned. Concurrently with the industry’s expansion, new challenges and risks surfaced, notably quality issues and safety concerns. To address these emerging issues, regulations were strengthened. This reflected a natural progression to rectify market failures and enhance consumer protection. While a subsequent adjustment phase of deregulation occurred, the initial regulatory focus predominantly emphasized risk-side aspects, primarily ensuring quality and safety.

The establishment of the FOSHU system, based on the scientific evaluation of food functionality, marked an epoch-making conceptualization of benefit-side regulation. Initially, FOSHU encompassed only foods in ordinary forms; however, it later extended its scope to include dietary supplements. Over time, regulatory adjustments for dietary supplements unfolded in distinct steps. First, risk-side deregulation occurred with the abolition of shape restrictions. Subsequently, benefit-side deregulation followed with the inclusion of dietary supplements within the FOSHU system. In a later phase, risk-side regulations were strengthened through the introduction of voluntary GMP, indicating a phased regulatory adjustment across both risk-side and benefit-side dimensions.

A recent significant development is the initiation of the FFC system. Compared to FOSHU, FFC labelling is permitted at a lower cost, signalling a further easing of benefit-side regulations. Conversely, quality control measures, particularly the promotion of GMP since around 2005 and the effective mandating of GMP for FFC dietary supplements in 2015, represent a strengthening of risk-side regulations. The CAA’s notification guidelines for FFCs [50] effectively mandated GMP for dietary supplements within the FFC system, reinforcing risk-side regulatory oversight. Thus, the FFC system can be viewed as a paradigm of benefit-side deregulation coupled with risk-side regulatory tightening. The evolution of regulation has, therefore, proceeded by dynamically balancing risk-side (quality and safety) and benefit-side (functionality information) considerations. A current review of FFC regulation is tightening both the risk side and the benefit side [49].

The market for functional foods developed before a complete regulatory system was in place. This early market emergence, combined with a layered regulatory approach that built upon existing rules, has created path dependency. This path dependency is a key factor shaping the complex functional food market in Japan. Path dependency refers to the self-reinforcing nature of earlier institutional choices that constrain future policy options [108].

### 4.2. Driving Industry by Balancing Risk-Side and Benefit-Side Regulations

Regulations are not merely passive adjustments to industry challenges that arise alongside market growth; rather, they can also actively drive industry development. Figure 2 conceptually models the dynamic mechanism by which regulation can drive industry development through the interplay of risk-side and benefit-side considerations. This section focuses on Japanese FFC regulations, which bundle risk-side and benefit-side considerations into an integrated framework. As Figure 2 illustrates, strategically linking elements like GMP on the risk side to opportunities like FFC health claims on the benefit side creates incentives.

The FFC system has enabled many companies, particularly smaller retailers, to enter the FHC market [17]. This deregulation has diversified the range of market participants, broadening the industry’s foundation. Moreover, the relatively high compound annual growth rates among companies utilizing FFC suggest that this scheme has helped drive their business expansion [17]. In contrast to the approval-based FOSHU system, the FFC system’s notification-based approach makes it harder for firms to gain a competitive edge solely by meeting regulatory requirements. Instead, its flexibility broadens strategic options, allowing companies to innovate and compete on their terms [18].

While deregulation on the benefit side has propelled the functional food industry’s growth, the simultaneous strengthening of risk-side regulations creates a ‘carrot-and-stick’ dynamic. For OEMs whose customers produce finished goods, there is a clear motivation to implement GMP. Doing so can expand their business and strengthen ties with client companies, especially those that actively use the functional food labelling system [16].

The FFC system, representing the benefit side of regulation, provides final product manufacturers with incentives to utilize functional claims, enabling them to offer products that are well-received by consumers and achieve higher sales [17]. On the other hand, the GMP system, representing the risk side of regulation, is closely linked to FFC. For OEMs, compliance with GMP entails short-term costs but offers long-term benefits by strengthening relationships with final product manufacturers, thereby creating incentives for GMP adoption [16]. This regulatory system leverages both risk-side and benefit-side regulations to maintain a balanced regulatory environment, fostering compliance among firms. Consumers benefit from enhanced safety and product quality, while both OEMs and final product manufacturers gain economic advantages. This dual regulatory system structurally and dynamically supports the development of the functional food industry. Through the linking of GMP to FFC, OEMs are incentivized to adopt GMP to reinforce partnerships with companies that use FFC. In turn, this dual introduction of FFC and GMP further promotes quality control. Essentially, the FFC system employs risk-side and benefit-side levers to maintain a balanced regulatory environment, fostering structural compliance among firms. Consumers benefit from enhanced safety and product quality, while both OEMs and finished-product manufacturers reap economic rewards. Consequently, this two-pronged regulatory system supports the functional food industry’s development structurally and dynamically.

As seen in the previous two sections, regulations do not simply develop in response to industrial challenges; they can also stimulate industry growth. In this way, the evolution of regulation and the expansion of industry proceed interactively and in parallel—a process best described as the co-evolution of regulation and industry.

Japan’s ageing, health-conscious population drives high demand; surveys indicate that a significant percentage of adults report purchasing health foods at least once a month or every few months [109]. Such consumer pull has repeatedly catalysed regulatory reform.

### 4.3. Lessons and Lens Based on Recent Health Incidents with Dietary Supplements

The 2024 Beni-koji (red yeast rice) contamination incident is a particularly instructive case for examining how regulatory frameworks, industry practices, and consumer safety intersect in the functional food sector. Tragically, the event involved serious health harm, underscoring the grave societal impact such cases can have. Analysing this incident through the lens of the dynamic interaction model presented in Figure 2 reveals how a failure in the risk-side management loop (specifically, the links from quality control/GMP and compliance incentive to product credibility) can disrupt the entire system and erode consumer trust. Government authorities mandated a nationwide product recall, initiated a far-reaching investigation into raw-material contamination, and collaborated with the National Institute on subsequent research. This series of events has spurred broader discussions on risk-side measures and quality management, leading to regulatory revisions of the entire FFC regulatory system, particularly focusing on strengthening the risk-side elements and feedback mechanisms illustrated in Figure 2. The long-term safety of functional ingredients had previously been ensured in the conventional FFC notification process by requiring the reporting of dietary experiences and results from safety studies. Furthermore, triggered by the recent Beni-koji incident, management against acute risks has been strengthened through enhanced manufacturing control via GMP and the reinforcement of the reporting system.

In the wake of this incident, some have pointed to flaws within the FFC system itself, including not only the absence of mandatory GMP certification but also insufficient mechanisms for third-party organizations to gather and disseminate information on health hazards, and inadequate transparency regarding the research papers underpinning health claims [110]. On the other hand, some point out that the FFC product-wide survey conducted by the CAA immediately after the incident would not have been possible for products outside the FFC system and that it demonstrated the usefulness of the notification-based FFC system [111]. As it is suggested that the direct cause is attributable to deficiencies in the manufacturing process, directly linking the issue to the flaws of the FFC system might be an overreach in terms of immediate causality. Nonetheless, given the historical trajectory of functional food regulation, it is understandable that discussions have broadened to encompass the system as a whole. Considering the longstanding balance between risk-side and benefit-side considerations, this regrettable juncture underscores a critical imperative to re-evaluate and refine the system. By framing the challenge as an impetus for reflection and systemic strengthening, we can strive for a more suitable regulatory framework—one that more effectively promotes both consumer health and the sustainable co-evolution of the industry and its regulation. Nevertheless, the complex nature of some functional ingredients and potential long-term health effects means that detecting all potential risks, particularly those with chronic or delayed manifestations, remains an ongoing challenge that requires the continuous evolution of regulatory science and post-market surveillance.

Taken together, this incident offers a sobering yet critical lesson and lens through which to study—and ultimately promote—the co-evolving relationship between regulatory policy, industry accountability, and public health protection in the realm of functional foods.

### 4.4. Interactions with Related and Surrounding Industries

The influence of regulations extends beyond the directly targeted functional food industry, impacting a broader ecosystem of related sectors. Conversely, the regulatory design of the functional food industry is shaped by the regulatory landscape of adjacent industries like pharmaceuticals and food, necessitating the consideration of these interactions.

#### 4.4.1. Impact of Functional Food Regulation on Related Industries

While this study centres on OEMs and end-product manufacturers, the regulations exert a broader influence, reaching into the diverse ecosystem of the functional food industry. These regulations create spillover effects throughout the supply chain, benefiting material manufacturers who can now develop FFC products [17]. The inclusion of fresh foods within the FFC system also empowers local businesses and stimulates the growth of regional industry [112]. Academia and researchers find expanded opportunities through industry collaborations, especially in providing scientific validation for functional claims [113,114]. The rising demand for clinical trial evidence may also involve medical doctors and related professions [50]. Consequently, the evolution of FFC regulation shapes the entire industrial ecosystem, extending beyond the directly regulated entities and strengthening firm–academia relationships.

#### 4.4.2. Influence of Regulations in Related Industries on Functional Food Regulation

On the other hand, the functional food industry is a convergence industry [16,28], influenced by adjacent sectors with differing regulatory levels, namely, pharmaceuticals and conventional foods, as shown in Figure 1. Functional food regulation is shaped by these related industries in both risk-side and benefit-side aspects, impacting elements of the dynamic interaction shown in Figure 2. Furthermore, regulation itself is also undergoing convergence, mirroring the broader industry trend. This is evident in how stricter pharmaceutical GMP regulations influence the ‘tightening risk-side regulation’ or ‘robust quality control’ components within the Figure 2 dynamic, incentivizing the transfer of pharmaceutical quality control expertise.

Risk-side regulations, such as GMP for functional foods, are notably influenced by the stricter pharmaceutical GMP regulations. This regulatory convergence is evident in the similarities between functional food and pharmaceutical GMP, signifying an erosion of regulatory boundaries. This converged regulatory design, in turn, drives a convergence of knowledge and technology, incentivizing the transfer of pharmaceutical quality control expertise to functional food manufacturing [16]. Benefit-side regulations, like the FFC system, are influenced by both food and pharmaceutical regulations. While drawing on food labelling guidelines, the FFC system also incorporates elements of pharmaceutical regulation in terms of efficacy evidence and advertising [18]. Designing functional food regulations requires taking into consideration the relative levels of and similarities to regulations in adjacent industries. Table 1 illustrates similarities between functional food and pharmaceutical regulations. These regulatory similarities and relative levels further facilitate industry convergence through knowledge spillover and firm entry. The regulatory gap between the medical and healthcare sectors impacts firm behaviour, incentivizing innovation in less-regulated areas while leveraging knowledge from stricter sectors.

In conclusion, effective functional food regulatory design necessitates a holistic perspective, considering interactions with related industries and the phenomenon of regulatory convergence as shown in Figure 1. Balancing risk-side and benefit-side regulations should account for the influences of and interdependencies with adjacent regulatory landscapes and recognize that regulation itself is a converging force, shaping both industry structure and firm behavior.

### 4.5. Prospects for the Global Market and Regulation

While this study focused on Japan’s functional food market, it is important to observe and compare environments in other regions and the characteristics of global companies, because the functional food market is expanding globally. As shown in Table 2, different countries have varying functional food regulations. Japan’s regulations are unique and complex in that they include multi-track health claim regulations (FFCs, FOSHUs, so-called health food). In contrast, the U.S. has a comprehensive DSHEA, covering labelling and GMP compliance. This strong and rigid regulation ensures consumer safety and promotes the growth of the dietary supplement industry.

In countries with emerging economies, the establishment of regulatory frameworks for functional foods is still insufficient. Japan is one of the world’s leading markets for functional foods, and recent regulatory reforms have attracted global attention. This study will also be of interest to policymakers in emerging economies that are building new regulations. For such economies embarking on the development of functional food markets, these findings underscore the importance of adhering to risk-side/benefit-side balanced regulatory design principles. In nascent markets, establishing a regulatory framework that carefully calibrates both risk-side and benefit-side considerations from the outset is crucial. This balanced approach can foster consumer trust and stimulate market growth while simultaneously ensuring product safety and quality.

### 4.6. Implications

#### 4.6.1. Implications for Policymakers and Administrators

For functional food regulation, policymakers should prioritize gradual system improvement while maintaining a balanced approach. The radical reform of the existing complex framework is unrealistic; instead, a steady focus on synchronizing benefit-side and risk-side regulations is advisable. Understanding the dynamic interplay illustrated in Figure 2 is crucial for designing effective regulations that leverage the links between risk management, compliance, market benefits, and innovation. Regulatory operations must remain flexible to accommodate diverse companies, thereby promoting both innovation and consumer protection. Policies should aim to strengthen the virtuous cycles shown in Figure 2, for example, by designing incentives that directly link compliance with risk-side requirements to market benefits and by fostering market conditions that stimulate innovation.

Building on lessons from recent incidents such as the Beni-koji case, several concrete directions are already under discussion. Authorities are designing a permanent mechanism for the continuous monitoring and ex-post evaluation of the FFC system, so that emerging scientific evidence or new social concerns can trigger timely, data-driven adjustments. In parallel, the large share of ‘so-called health foods’ that still fall outside any functional food framework highlights the need for a broader review of the entire regulatory perimeter. Expanding oversight—proportionately to risk—would close protection gaps and align incentives across all market segments.

Enhancing public health requires open, ongoing dialogue with a wide range of stakeholders. The system should integrate insights from health economics, industrial promotion, and evidence-based policymaking. Key initiatives include multifaceted collaboration among consumers, companies, industry associations, and experts, as well as sharing Japan’s regulatory experiences internationally, in line with global trends. Industry associations, in particular, play a pivotal role in this process.

#### 4.6.2. Implications for Companies and Industry Associations

Industry associations should leverage their broad membership base to develop constructive, industry-wide policy recommendations. In the food sector, numerous types of companies coexist—from large corporations to small and medium-sized enterprises, and from manufacturers to distributors and retailers. Because each firm has its own distinct interests, industry associations must consolidate these varied perspectives and establish frameworks, such as consortia, that discourage opportunistic behaviour and uphold long-term sustainability.

In Japan, the creation of a Fair Competition Code for functional foods is now underway. Because such self-regulation operates outside formal statutes yet can shape marketing practices, product quality, and consumer trust, it represents a critical but under-studied layer of governance. Future work should quantify how these private codes interact with, and potentially reinforce, statutory requirements.

Establishing voluntary standards and guidelines can effectively complement formal regulations. By combining flexible self-regulation with legal frameworks, industry and government can harmonize regulatory oversight with industrial growth, ultimately fostering a sustainable market and enhancing consumer welfare. Industry associations could also play a vital role in harmonizing voluntary standards across the supply chain. Through the sharing of data on product efficacy, safety, and labelling, associations can help companies coordinate their R&D investments.

From the perspective of mitigating information asymmetry, industry associations can significantly bolster consumer awareness and trust. One effective strategy involves creating guidelines to enhance labelling clarity, ensuring that product information, particularly regarding health benefits, proper usage, and potential risks, remains both accurate and comprehensible. Additionally, consumer education initiatives—such as workshops, informational campaigns, or partnerships with advocacy groups—can further empower individuals to make more informed decisions. By actively leading these efforts, industry associations not only help consumers navigate the functional food market with confidence but also cultivate a more transparent and responsible industry landscape.

### 4.7. Limitations and Future Perspectives

This study has limitations in several aspects. Firstly, the focus on a single case study, the Japanese functional food system, restricts the generalizability of findings to other regulatory contexts. Secondly, the reliance on a literature review and publicly available data limits the empirical depth and robustness of the analysis, lacking primary data from firms and consumers. Thirdly, the risk-side/benefit-side framework, while analytically useful, may oversimplify the complexity of real-world regulatory systems. Fourthly, this study’s focus on functional foods necessitates caution when extrapolating conclusions to other sectors within or beyond the healthcare industry. Lastly, the predominantly industry-centric perspective may not fully capture the diverse viewpoints of consumers and other stakeholders. Future research should address these limitations by expanding the scope geographically, incorporating primary data and quantitative analysis, refining the framework, exploring broader industry applications, and integrating multi-stakeholder perspectives.

This study focuses on the Japanese functional food sector, yet the risk-side/benefit-side framework could be applied to a variety of contexts. Future research might conduct cross-country comparisons to examine whether similar regulatory co-evolution patterns appear in other markets, especially emerging economies where health food systems are rapidly evolving but remain less formally regulated. Additionally, as healthcare convergence advances, exploring newer product categories would test the framework’s versatility. Employing quantitative event-study methodologies or multi-stakeholder interview approaches could also deepen insights into how regulatory shifts affect firm entry, product innovation, and consumer perceptions. Such comparative or expanded research efforts would not only validate and refine the risk-side/benefit-side framework but also provide broader lessons for regulatory design in other convergence industries, contributing further to the fields of innovation management and regulatory science.

## 5. Conclusions

This study proposed and validated the risk-side/benefit-side framework to systematically analyse the roles and impacts of functional food regulations, supported by a literature review and a case study of Japan’s functional food industry. The framework highlights the complementary and balanced evolution of risk-side (quality and safety) and benefit-side (functionality) regulations, which are crucial for the industry’s development.

The present study’s core academic contribution lies in elucidating the co-evolutionary structure of regulation and industry through the lens of regulatory duality. Moving beyond a static view of risk-side and benefit-side regulations as a simple trade-off or binary opposition, this study proposes a dynamic understanding of their relationship, suggesting a process akin to sublation. As conceptually illustrated in Figure 2, this framework highlights how robust risk-side measures (e.g., mandatory GMP), while initially appearing as compliance costs, can paradoxically generate significant benefits for firms (e.g., enhanced reputation, operational efficiency, access to specific market segments like FFCs through linkage). This corporate benefit, in turn, contributes to consumer welfare in a layered manner, not only by ensuring product safety but also by fostering trust and enabling informed choices regarding functionality. Thus, the framework presents a meta-level discussion where risk-side and benefit-side considerations are not in conflict but rather cooperate and circulate in a virtuous cycle that drives value creation.

The Japanese case study provides practical insights into how functional food regulations have co-evolved with the industry, vividly demonstrating this dynamic interplay. Initially, risk-side regulations prioritized safety and quality standards. Later stages introduced benefit-side mechanisms, such as the FFC system, linking them implicitly or explicitly with enhanced risk-side requirements, to address information asymmetry and foster innovation. This phased and dynamic balancing act reveals how regulatory design can leverage the symbiotic relationship between managing risks and promoting benefits to both foster industry growth and ensure robust consumer protection. This offers lessons for other countries, especially emerging economies, in designing regulations that balance consumer protection and industry growth.

By integrating insights from social and economic regulation, the risk-side/benefit-side framework complements existing theories like the Porter Hypothesis (by explaining how regulations can drive innovation through risk management incentives) and regulatory capture theory (by suggesting how balancing both dimensions can mitigate capture risks). It offers a tailored approach for healthcare convergence industries where both scientific evidence and public trust are paramount.

Key findings include the following:Risk- and benefit-side framework as sublation: effectively reveals the dynamic, symbiotic, and value-creating relationship between regulation aimed at consumer protection (risk side) and innovation/information (benefit side), transcending a simple trade-off and highlighting how risk management can drive benefits.Co-evolution driven by dynamic linkage: demonstrates how Japanese regulations have dynamically linked market access and innovation incentives (benefit side) with safety and quality assurance (risk side), illustrating how strengthening one dimension can positively reinforce the other and drive industry development.Analytical effectiveness: provides a structured tool to understand the complex dynamics of functional food regulations, emphasizing their dual, adaptive, and mutually reinforcing nature.

This study not only advances regulatory science by offering a novel theoretical lens for understanding the dynamic interaction between risk and benefit in regulation but also offers practical guidance for policymakers and industry stakeholders. By highlighting how balanced and integrated regulatory design can create a virtuous cycle of risk management leading to benefit enhancement, it contributes to theory development in the healthcare sector and provides a foundation for future research. The Japanese case serves as a valuable empirical example of how such balanced regulatory design can drive both industry growth and consumer trust, making this study relevant for academics and practitioners alike.

Future research could explore this dynamic interplay and the sublation process in other convergence industries or regulatory contexts where balancing risk and benefit is critical. Quantitative studies examining the impact of specific regulatory linkages (e.g., the effect of mandatory GMP adoption on firm performance and market share under FFC) would further validate this framework. Investigating consumer perception regarding the link between stringent safety standards and trust in functional benefits also presents a valuable avenue. Furthermore, exploring the role of international regulatory harmonization in facilitating or hindering this dynamic could provide valuable insights for a global market.

## Figures and Tables

**Figure 1 foods-14-01581-f001:**
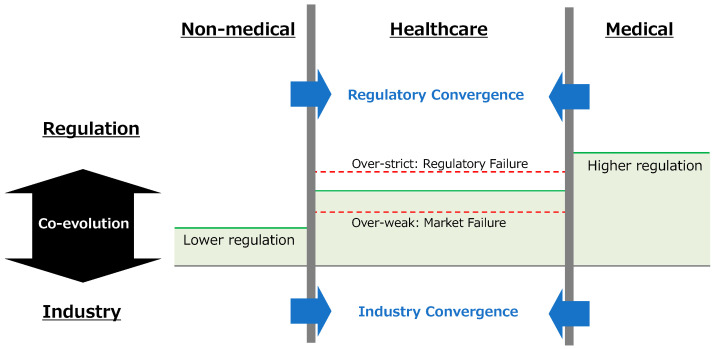
Regulatory and industry convergence in healthcare sectors. For a detailed explanation, please refer to Section 2.1 in the text.

**Figure 2 foods-14-01581-f002:**
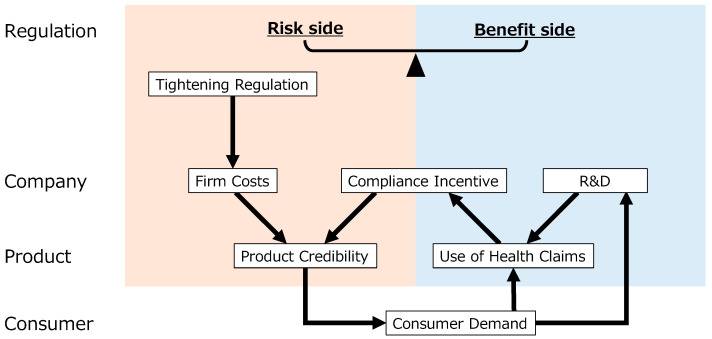
Dynamic interaction mechanism between risk-side and benefit-side considerations. For a detailed explanation, please refer to Section 2.4 in the text.

**Table 1 foods-14-01581-t001:** Comparison between functional foods (so-called health food and food under the FHC regulatory system) and pharmaceuticals.

Category	So-Called Health Food	Foods with Health Claims (FHCs)			Pharmaceutical
		Foods with Function Claims (FFCs)		Foods for Specified Health Uses (FOSHUs)	
Administrative process	None	Notification		Approval	Approval
Basis of efficacy or functionality	Not allowed to label functions	Systematic review (SR) of functional components (research on clinical trials)	Product clinical trials	Product clinical trials (large-scale)	Nonclinical studies, clinical trials (3 phases)
Basis of safety	Not required (eating experience assuming safety)	Acceptable by food experience (if the basis of dietary experience is insufficient, it will be based on clinical trials, etc.)		Clinical trials	Nonclinical studies, clinical trials
GMPs as quality control regulations for dietary supplements	Not mandatory (voluntary GMPs)	GMPs will become mandatory from September 2026		Not mandatory (voluntary GMPs)	GMPs are mandatory (high-level)
Market size [JPY billion]	1622 (2015) 1645 (2020)	31 (2015) 357 (2020)		378 (2015) 321 (2020)	-

Note: Foods with Nutrient Function Claims (FNFCs) are omitted from the table because the system only allows the labelling of their functions in the case of nutrients specified in the standard specifications. GMPs: Good Manufacturing Practices.

**Table 2 foods-14-01581-t002:** Comparison of functional food regulatory systems in various countries.

	EU	US	Korea	Taiwan	China	Singapore	Japan
Registration/Approval System (Business Item)	Yes	No	Yes	Yes	Yes	No	In parallel
GMP System by Country for Dietary Supplements	Yes	Yes	Yes	Yes	Yes	No	Yes
Clinical Trials of Individual Products	For new ingredient	Not required	Depends	Depends	For new ingredient	Not required	Depends

**Table 3 foods-14-01581-t003:** List of the literature on functional food regulations from the perspectives of the risk side (quality and safety) and benefit side (functionality), focusing on either consumers or companies.

	Regulation	
	Risk side Protection from risk (quality, safety)	Benefit side Information about the benefit (functionality)
Consumer	- Quality assurance, safety (Pravst 2011 [64], Costa 2012 [59], He 2015 [56], da Cruz 2006 [57], Starr 2015 [13], Dickinson 2011 [54], Van Breemen 2015 [55], Sato 2020 [16]) - Credibility (Sääksjärvi 2009 [75])	- Inform of health benefits by health claim (Bornkessel 2014 [76], Tarabella 2012 [77], Sato 2023 [17]) - Motivation for purchase (Sääksjärvi 2009 [75]) - Protection from inappropriate information
Company	- Burden of compliance cost (Tunalioglu 2012 [70], Escanciano 2014 [71]) - Quality/safety for product (Costa 2012 [59], He 2015 [56], Sato 2020 [16]) - Increasing quality and safety - Decreasing consumer complaints - Internal effect on company (da Cruz 2006 [57]). - Improving productivity - Motivating/educating employees - Effect for marketing (Sampaio 2009 [72], Crowley 2006 [73], Holleran 1999 [74], Sato 2020 [16]) - Building relationships with customers - Promoting product marketing	- Burden of R&D cost/administrative procedure cost (Farid 2019 [25], de Boer 2015 [79], Sato 2024 [18]) - Improvement in product value (Sääksjärvi 2009 [75], Bornkessel 2014 [76], Tarabella 2012 [77], de Boer 2015 [79], Hobbs 2014 [63], Sato 2023 [17]) - Promotion by health claims - Development of technology for innovation - Differentiation other than patent - Providing appropriate competitive environment (Hobbs 2014 [63], Santini 2018 [62], Herath 2008 [81], Lalor 2011 [80], Chauhan 2013 [61], Sato 2023 [17]) - Activating industry - Appropriate market competition

**Table 4 foods-14-01581-t004:** The regulatory changes on the risk side (safety or quality control) and the benefit side (health claims) in functional foods in Japan.

Decade	Event	Related Regulation	Risk-Side	Benefit-Side
Before 1960s				
1960s	Rise of the functional food market.	-	-	-
1970s	Dietary supplements with pharmaceutical-like shapes were prohibited.	‘46 Notice’ (1971)	Tightening	
1980s	Relaxation of shape restrictions. Must be labelled ‘food’.	Revision of ‘46 Notice’ (1987)	Deregulation	No concept (Prohibited)
1990s	FOSHU started, but dietary supplements were not allowed in FOSHU.	Foods with Health Claims (1991)		Deregulation (Clarification of rules)
	Dietary supplements with pharmaceutical-like shapes were allowed.	Revision of ‘46 Notice’ (1997~2000)	Deregulation	
2000s	Dietary supplements such as FOSHUs were allowed.	Revision of the Health Claims Food System (2001)		Deregulation
	Introduction of voluntary GMP for dietary supplement products.	GMP Certification (2005)	Tightening	
2010s	GMP became practically mandatory for dietary supplements. The cost of health claims decreased with the SR/certification system.	FFC System (2015)	Tightening	Deregulation
2020s	GMP became mandatory for dietary supplements. Improvement in the accuracy of health claims and visibility of warning messages on product packages.	FFC System (2024)	Tightening	Tightening

## Data Availability

No new data were created or analysed in this study.

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
