# Peer review of "The Co-Evolution of Markets and Regulation in the Japanese Functional Food Industry: Balancing Risk and Benefit"

_foods, 2025, doi:10.3390/foods14091581_

Round 1
Reviewer 1 Report
Comments and Suggestions for Authors
Dear Authors,
The review with title "o-evolution of Markets and Regulation in the Japanese Functional Food Industry: Balancing Risk and Benefit" is very interesting because it deals with the most popular topic "functional food". Furthermore, the regulations are very important due to the fact that functional food should be safe, acceptable and to have positive impact to the human health. However, this review might be improved by several points:
- The authors should include and discussed the regulations which makes clear separation of native and non-native functional food.
- The regulations which included fortification of the functional food should be included (doi: 10.9745/GHSP-D-15-00171)
- Important information for functional food regulative can be found in link https://www.google.com/url?sa=t&rct=j&q=&esrc=s&source=web&cd=&ved=2ahUKEwiy2KjrnsiMAxXwQvEDHVcBFQ4QFnoECBYQAQ&url=https%3A%2F%2Fone.oecd.org%2Fdocument%2FGOV%2FRPC(2024)4%2FANN1%2Fen%2Fpdf&usg=AOvVaw3574jvTb9wNNL-HjIRWRCH&opi=89978449
- I suggest the discussion of the manuscript (doi: 10.3390/nu13041118)
- Line 617-622: The authors should include reference of the links or regulations for minimal medicinal impact of certain food to be classified as "functional".
- The authors should mention of the most common frauds when certain "food" is classified as "functional".
I suggest major revision of the review.
Reviewer 2 Report
Comments and Suggestions for Authors
This paper provides a systematic analysis of the co-evolution of regulatory frameworks and market development in Japan’s functional food industry, proposing the innovative “Risk-side and Benefit-side Framework” to examine the dynamic balance of functional food regulation. The research topic aligns withe global trends in the healthy industry, and the paper is well-structured with rich data sources, making it valuable for both policymakers and academia. However, there is still room for improvement in some aspects of the article.
- Introduction: This section should clearly outline the background of the functional food market, including global growth trends, changes in consumer demand, and regulatory challenges, to enhance the motivation and significance of the research presented in the article.
- Lines 154 to 159: Functional food regulatory policies in different countries and regions should be compared in detail, with particular emphasis on the uniqueness and complexity of the Japanese regulatory system.
- Section 2.6: "risk-side and benefit-side frameworks" relies on conceptual division. It is suggested to further elucidate the dynamic interaction mechanism between them.
- The schematic in Figure 1 is text-intensive, improve the quality of the diagram.
- Section 4.6.1: Please propose a specific action plan.
- Conclusion: Please outline future research directions and provide more practical policy recommendations.
Reviewer 3 Report
Comments and Suggestions for Authors
This is a well designed, well written “case study” of a leading country’s balance between risk and benefit
Line (L) 34 Consider some supporting data (facts) to back up your claims
L 49—There is a ref for this statement, unless you think you invented it?
L 66 Consider “are continuing to evolve …
Table 1 looks good, but I’m left wondering what about the rate of change in value of Yen or some other measure over a 5, 10 year time frame to provide the reader with some perspective
152 –links to those regulatory agencies specifics on this topic would be beneficial
190 I’d add allergenistic issues as one of the major concerns
224 I’m not following the point you are trying to make, please re word
Table 3 good concise synopsis!
329 Great intro—one of the questions from industry has been, so “how do you know” many regulations are based on the precautionary principle an no data
L 382—Typo on year
L 416 You are doing an excellent job tracing the historical development. Pl include the FDA’s decision to not regulate under DESHA—basically (because of Congressional pressure) waiting for the shoe to drop or a group of consumers to suffer harm before FDA would get involved
L 440 Again, data supporting your “expanded rapidly” would help
Table 4 provides a good thumbnail summary
L 532 Are you going to clarify “path dependency” ?
L 572 Good point—one question keeps coming up, you have apparently chosen not to include Consumer Demand? Japan is viewed as a place of fervent demand for health enhancing “foods”. Can you include a paragraph speaking to this?
L 583 True for acute toxicity, but many of these mold aflatoxins are more chronic toxicants. What is the mechanism to detect a potential toxin that significantly increases the risk of cancer or other dire health consequences—in the long term?
Fig 1—not sure this adds much to your rich discussion
L 704 Perhaps data for a follow-on article is industry self-regulation in this space.
Reviewer 4 Report
Comments and Suggestions for Authors
The paper ‘Co-evolution of Markets and Regulation in the Japanese Functional Food Industry: Balancing Risk and Benefit’ tackles an interesting research topic. The authors focused their attention on the issue of functional foods, the importance of which has evolved beyond essential nutrition to assume an increasingly strong importance of a preventive nature. With this in mind, the authors focused their attention on analysing the regulatory dimension in this field, proposing a framework for a systematic review in relation to the risk (quality and safety of use) and benefit system (functionality) of functional foods. For the research, a historical analysis of the regulatory landscape of functional foods in Japan over the period 1960-2025 was performed to analyse the evolution of the regulatory regimes, taking into account their impact on the development of the industry.
The abstract of the paper presents the reference and scope of the research, but without explicitly specifying the purpose of the research, which needs to be completed. Furthermore, the background is outlined without articulating the research gap to which this review responds (signalled in the introduction), which also needs to be supplemented.
The introduction raises issues presenting the background to the research, pointing out the problem of functional foods, presenting their essence and role in therapeutic processes. Here, the authors point out the problem of the lack of a clear regulatory framework (with reference to sources), indicating the research questions (section 1.2), justifying the research topic adopted. Section 1.3 presents Japan's functional food fegulation, with a discussion required under the table with a summary of the regulations (line 105).
Section 1.4 of the introduction presents the structure of the study. This section should be expanded on the research steps and the review methodology adopted. I propose to address the above in a separate section on ‘materials and methods’ and discuss the data sources, their quantity, mode of acquisition and processing. At this point it would be useful to include a research model for this review, including the research steps with an attribution of the methodology used. In this regard, I make comments to improve the content of the introduction and refine the structure of the paper by isolating the noted structure element - the ‘materials and methods’ section.
Another element of the study is the literature review. The review is a valuable comprehensive study, but it should be oriented towards justifying the research threads adopted by demonstrating a gap in the literature. In this respect, the introductory content should be linked to the research threads presented in the introduction, adopted for this review.
In addition, research threads (such as the zero-sum assessment of the existence of regulations in the area under study in other countries (research method), should be moved to the area ‘results of the review’. In this respect, I suggest structuring the content of the literature review, which is a valuable element of this study.
The next steps are oriented towards the analysis of the regulatory system for functional foods (item 3), followed by an exploration of the regulatory schemes for the industrial system in this area (item 4). In the area of point 4, a discussion of the content of the figure is required under its name rather than in the name (lines 659-669).
The summary of the paper is very short, I believe it needs to be expanded. The summary should more strongly emphasise the contribution of the study to the existing literature in response to the diagnosed gap. I suggest improving the above.
Literature well chosen, but it is worth strengthening the scope.
In conclusion. Paper needs improvement prior to publication. Improvement is needed in the abstract, introduction, literature review and conclusion, to the extent indicated in the review. In addition, minor additions are needed in points 3 and 4 - according to the suggestions indicated. Furthermore, the structure of the paper needs to be improved according to the content of the review.
Round 2
Reviewer 1 Report
Comments and Suggestions for Authors
Dear Authors,
The revised version of the review with title "Co-evolution of Markets and Regulation in the Japanese Functional Food Industry: Balancing Risk and Benefit" is significantly improved and all my suggestion and comments as well as addition of three references are corrected and explained well. Furthermore, the corrected Table 4 and Figure 2 are more self-explaining and very clear.
I suggest acceptance of the review in this revised version.
Reviewer 4 Report
Comments and Suggestions for Authors
The authors have improved the article.